# The Performance of the Magneto-Impedance Effect for the Detection of Superparamagnetic Particles

**DOI:** 10.3390/s20071961

**Published:** 2020-03-31

**Authors:** Alfredo García-Arribas

**Affiliations:** 1Departamento de Electricidad y Electrónica, Universidad del País Vasco UPV/EHU, 48940 Leioa, Spain; alfredo.garcia@ehu.es; 2Basque Centre for Materials, Applications and Nanostructures, BCMaterials, 48940 Leioa, Spain

**Keywords:** magneto-impedance, biosensor, finite-element method

## Abstract

The performance of magneto-impedance sensors to detect the presence and concentration of magnetic nanoparticles is investigated, using finite element calculations to directly solve Maxwell’s equations. In the case of superparamagnetic particles that are not sufficiently magnetized by an external field, it is assumed that the sensitivity of the magneto-impedance sensor to the presence of magnetic nanoparticles comes from the influence of their magnetic permeability on the sensor impedance, and not from the stray magnetic field that the particles produce. The results obtained not only justify this hypothesis, but also provide an explanation for the discrepancies found in the literature about the response of magneto-impedance sensors to the presence of magnetic nanoparticles, where some authors report an increasing magneto-impedance signal when the concentration of magnetic nanoparticles is increased, while others report a decreasing tendency. Additionally, it is demonstrated that sensors with lower magneto-impedance response display larger sensitivities to the presence of magnetic nanoparticles, indicating that the use of plain, nonmagnetic conductors as sensing materials can be beneficial, at least in the case of superparamagnetic particles insufficiently magnetized in an external magnetic field.

## 1. Introduction

Magnetic nanoparticles (MNPs) are used in numerous biomedical applications, both for diagnosis and therapy [1]. In diagnosis, they are used for magnetic resonance imaging enhancement [2] and, after adequate functionalization, for magnetic separation, concentration, and detection of specific analytes [3]. In therapy, MNPs are used in drug delivery and for hyperthermia treatments, among others [4]. The most relevant type of magnetic particles for these applications are superparamagnetic iron oxide nanoparticles, SPIONs. In order to detect the presence of MNPs and to quantify their concentration, different types of magnetic sensors have been proposed [5,6]. In particular, the magneto-impedance (MI) effect’s extraordinary sensitivity to small magnetic fields has motivated the active development of MI-based biosensors which, fundamentally, detect the presence and concentration of magnetic particles [7,8,9,10,11,12,13].

The MI effect consists in the large variation of the electrical impedance Z of a soft magnetic conductor when subjected to an external magnetic field [14]. The phenomenon can be understood from the classical electromagnetic theory as a consequence of the skin effect’s dependence on the permeability *μ* of the material. The skin effect—that is, the limited penetration of the alternating electromagnetic field in a conductor, is characterized by the penetration depth *δ* given by
(1)$$\delta ={\left(\pi f\sigma \mu \right)}^{-1/2}$$
where *f* is the frequency of the field, and *σ* the conductivity of the material. Depending on the magnetic behavior of the sample, the external magnetic field modifies the permeability, which subsequently changes the effective electromagnetic cross section of the conductor, and produces concomitant variations in the impedance. Figure 1 shows a sketch of a typical MI curve in a planar sample with transverse anisotropy. The maximum impedance *Z*_max_ is obtained for an applied field similar in magnitude to the anisotropy field *H^k^*, when the transverse permeability reaches its maximum value *µ*_max_. The minimum impedance *Z*_min_ is obtained when the sample is magnetically saturated and the permeability *µ*_min_ is close to *µ*_0_.

In this kind of sample, the largest sensitivity to the external magnetic field is reached in the region between *H* = 0 and at the peak where *H* = *H^k^*. For instance, the sensitivity of a multilayered planar sample can reach a value of 27 kΩ/T, occurring at an applied field of *µ*_0_*H* = 200 µT (*H* = 180 A/m) and measured at 23 MHz [15].

The MI is usually quantified as the ratio of impedance change, and its maximum value (at each frequency), given by [14]
(2)$${\mathrm{MI}}_{\max}\left(\%\right)=\frac{Z_{\max}-Z_{\min}}{Z_{\min}}\times 100$$
is usually used as a figure of merit for MI sensors. In the literature, when used for detecting the presence and concentration of MNPs, it is usual to report the changes produced by the presence of nanoparticles in the complete MI curve and in particular, in its maximum value MI_max_. Wang and collaborators [16] have compiled the results from quite a number of works that use the MI effect to detect magnetic particles. Overall, the comparison of the results seems to be utterly inconsistent, because some authors report an increase in the MI response when increasing the concentration of particles (for example, Devkota and collaborators [17]), while others describe a decreasing tendency (for example, Yang and collaborators [18]). As the MI materials and the nature of the particles differ between studies, it becomes necessary to adopt a systematic approach to evaluate the sensitivity of the MI effect to quantify the concentration of MNPs. The present work attempts to shed light on this issue from a theoretical point of view, using a numerical procedure (finite element method) to solve Maxwell’s equations and calculate the impedance in the conditions of the experiments described in the literature.

An important point is that the authors of the mentioned works compiled in [16], in which contradicting results are reported, usually do not provide a satisfactory explanation regarding the physical origin of the MI effect’s sensitivity to the presence of magnetic nanoparticles. It is assumed that it is a consequence of the MI effect’s extraordinary sensitivity to low magnetic fields. The fact, however, is that superparamagnetic particles do not display remanence, and therefore do not produce external magnetic fields unless magnetized by an appropriate biasing field. However, a large applied magnetic field saturates the MI sensor and drastically reduces its sensitivity. In these circumstances, the sensitivity of the MI sensor to the presence and concentration of the particles cannot be caused by the magnetic field that they produce. Of course, this is true only if the particle system shows true superparamagnetic behavior. Agglomeration, or a large particle-size distribution, can produce a finite remanence which affects the MI signal of the sensor. The analysis of experimental data becomes difficult without a detailed particle characterization. There are some works in which the MI effect has been properly used to quantify the concentration of superparamagnetic nanoparticles in a clever configuration by measuring their stray field when magnetized by a strong external field [10], but in most of the works that claim the detection of magnetic nanoparticles using the MI effect, there is no satisfactory explanation of the involved mechanism.

In this work, we make the more realistic assumption that it is the permeability of the magnetic nanoparticles that produces the change in the MI response of the sensor. The sensing mechanism is therefore very simple: the presence of a high permeability medium in the proximity of the MI sensor modifies the distribution of the electromagnetic field associated with the alternating current flowing in the sensor. This produces a variation of the sensor’s impedance, but it is not related to the intrinsic sensitivity of the MI to low magnetic fields.

As explained in the next section, this concept is implemented in this work in the calculation of the MI response. The results from the numerical solution of Maxwell’s equations not only confirm that our assumption qualitatively reproduces the experimental results, but also help to explain the discrepancies reported in the literature compiled in [16], about the MI behavior when detecting magnetic nanoparticles.

## 2. Numerical Calculation Procedure

We aim to evaluate the response of a magneto-impedance sensor in the presence of a sample containing superparamagnetic particles. The electrical impedance of a conductor can in principle, be analytically calculated using Maxwell equations. In practice, only in very simple cases with highly symmetric geometries can a closed expression be derived, usually with the use of severe approximations. For instance, in planar samples, infinite width and length is assumed to calculate the impedance, producing an expression as [19]:(3)$$Z=R_{dc}\sqrt{j}\theta \cot h\sqrt{j}\theta$$
for a sample of thickness 2*a*, where *R_dc_* is the *dc* resistance, $j=\sqrt{-1}$, the imaginary unit, and $\theta =a\sqrt{2\pi f\sigma \mu }=\sqrt{2}a/\delta$. Similar expressions are obtained for the case of cylindrical samples (wires).

Using this type of expression, the magneto-impedance curve can be computed by incorporating the field dependence of the permeability, using models for the magnetization process and the dynamical behavior of magnetization [20,21]. Although useful, these models systematically overestimate the magnitude of the MI effect [22,23] when compared with actual measurements, mainly because the real conditions of the experimental setup are not usually considered. In particular, the contribution of the measuring circuit to the total impedance is not usually considered in the models.

Numerical simulation using Finite Element Methods (FEM) can overcome these limitations as the elements of the measuring circuit can be easily incorporated in the model. After specifying the material properties and the adequate boundary conditions, the impedance can be calculated by numerically solving Maxwell’s equations for an arbitrary geometry. If the dependence of the material properties, especially the permeability, on the magnetic field *H* are known, the complete *Z*(*H*) can be calculated. If only the performance (expressed as the MI ratio in Equation (2)) is of interest, FEM can be used to calculate the impedance for two values of permeability: *µ*_max_, corresponding to the maximum value of transverse permeability (and the impedance *Z*_max_) and *µ*_min_ = *µ*_0_, corresponding to the saturated state (and the impedance *Z*_min_) as depicted in Figure 1.

The versatility of FEM makes it possible to calculate the impedance of the MI sample in a variety of environments accurately resembling the experimental conditions. For the purpose of this work, using FEM, we calculate the impedance of an MI material in the presence of a sample of magnetic nanoparticles which is modeled as a homogeneous continuum material with a given permeability *µ_P_*. As explained before, the permeability of the particles is the only relevant magnetic property in this approach. There is no attempt to model the individual magnetic behavior of the particles. To reproduce the conditions of the experiments found in the literature, the MI response is evaluated as a function of the concentration of the nanoparticles. If we assume that the nature of the particles is always the same, the mean permeability of the sample of magnetic nanoparticles is simply proportional to its concentration. That is, we can calculate the evolution of the MI sensor’s response in the presence of a sample of magnetic nanoparticles with different concentrations by simply modifying the value of *µ_P_*—the permeability of the MNPs system.

For the numerical calculations, the free 2D (two-dimensional) software package FEMM [24] is used. Its implementation is restricted to low-frequency electromagnetic problems, ignoring the displacement current. Therefore, it cannot simulate propagating effects or account for the dielectric properties of the materials. However, it can accurately resolve MI problems in which the important effects are the skin effect and the magneto-inductive effect [25].

As previously discussed, the contribution from the measuring circuit to the impedance must be considered to obtain realistic results. For that reason, the simulation is performed with the MI sample inserted in a microstrip transmission line, which is a popular text fixture for measuring MI in planar strips. Figure 2a schematizes the setup of the simulated experiment. The sample of magnetic nanoparticles is placed in the shape of a drop on the MI sensor in the microstrip transmission line. The sketch Figure 2b illustrates the layout of the 2D finite element problem. The simulation domain represents the middle plane of the real problem. Due to the symmetry, only half of it needs to be simulated. The simulation domain (including the space surrounding the microstrip line) is kept small to reduce computation time. Built-in FEM open boundary conditions are imposed on the boundary to guarantee correctness of the solution.

The drop of magnetic nanoparticles is modeled as a nonconducting, homogeneous continuum material with a given permeability *µ_P_*. As previously discussed, increasing values of *µ_P_* correspond to increasing concentrations of MNPs.

The MI sample is modeled as a homogeneous conductor (*σ* = 6.6 × 10^5^ S/m) with a constant permeability *µ*. No definite magnetization process is assumed, and the MI is calculated as the relative ratio between the values of the impedance obtained with a high value of permeability *µ*_max_ and with a value of *µ*_min_ = *µ*_0_. The geometry of the MI sample is kept constant through all the simulations: 1 mm wide and 20 µm thick (the length is not relevant for the 2D problem, and the net value of the impedance is calculated for a sample 1 m long).

In the microstrip line, the ground plane is modeled from pure cooper (*σ* = 5.8 × 10^7^ S/m, *µ* = *µ*_0_) with a thickness of 35 µm. The 0.8 mm thick dielectric presents no conductivity, and *µ* = *µ*_0_.

To calculate the impedance, an alternating current of a given frequency is imposed to flow through the sample, perpendicular to the plane of simulation. The current returns in the opposite direction through the ground conductor.

In this work, the MI is calculated for a large number of configurations with different values of *µ*_max_ and *µ_P_*. For each configuration, the MI is calculated as a function of the frequency, in a range from 0 to 150 MHz or 0 to 1 GHz, depending on the case. To accommodate the intensive computational resources needed, we have made use of the XFEMM implementation of the software [26], which is run in a computer cluster.

## 3. Results and Discussion

Let us consider first the case of the MI sensor without any MNPs. Figure 3 presents the calculated MI response as a function of the frequency for a sample 1 mm wide and 20 µm thick. With the value of *µ*_max_ = 5000 *µ*_0_ used in the simulation, the maximum value of MI ratio is MI_max_ = 568%. The inset in Figure 3 shows the experimentally measured MI ratio of an amorphous ribbon composed of Co_65_Fe_4_Ni_2_Si_15_B_14_. We can observe that the shapes of both curves are essentially similar, although the MI experimental values are significantly lower, indicating that the permeability *µ*_max_ = 5000 *µ*_0_ used in the simulation is largely overestimated. Nevertheless, this case, which corresponds to a very sensitive MI sensor, is considered the starting point in our goal of studying the relevance of MI to detect magnetic nanoparticles.

When a drop containing MNPs is placed on top of the MI sensor as described in Figure 2, the complete MI curve as a function of the frequency is modified. Figure 4 shows the variation of the region of the MI curves around the maximum when the concentration of MNPs is monotonically increased (increasing values of *µ_P_*). The simulation results are divided into two plots for better clarity. Figure 4a determines that the MI response decreases when the permeability of the MNPs system increases from 2 *µ*_0_ to 15 *µ*_0_. However, when *µ_P_* is increased further, Figure 4b shows that the MI response changes tendency and start to increase. In both plots, the large red and black dots indicate the position of MI_max_ for the different values of *µ_P_*. Figure 5 compiles these results, plotting the evolution of MI_max_ for the whole range of tested *µ_P_* values.

Figure 5 shows that depending on the value of the permeability of the system of MNPs, the sensitivity of a very sensitive MI sensor can have a negative or positive value. That is, the MI ratio can display either a decreasing (for *µ_P_* < 15) or an increasing (for *µ_P_* > 20) behavior. This can certainly explain the discrepancies found in the compiled works [16] indicated in the introduction: the situation in the works where a decrease in the MI ratio is reported when increasing the concentration of MNPs, according to the results shown in Figure 5, can correspond to situations in which the permeability of the nanoparticle ensemble is low (due to a low intrinsic permeability of the nanoparticles or low concentrations). In contrast, the situation in the works reporting an increasing MI response with concentration can correspond to cases in which the permeability of the MNPs system is large.

As the maximum impedance ratio MI_max_ is calculated as a quotient as displayed in Equation (2), the negative slope of the curve in Figure 5 for *µ_P_* < 15 indicates that *Z*_min_ increases more than (*Z*_max_ − *Z*_min_) in this range. This is easily explained considering that the increase in the impedance is due to the presence of a magnetic medium (the particles) near the MI conductor. It is the same phenomenon occurring when a soft ferrite increases the impedance of a wire in a RF choke. When the permeability of the medium is very low compared with the permeability of the conductor itself, as in *Z*_max_, the influence of the magnetic medium is low. However, when the permeability of the conductor is low, as in *Z*_min_, even a surrounding medium with a low permeability produces a change in the impedance.

It is expected that the intrinsic MI performance of the sensor must have an influence on its sensitivity to the presence on MNPs. The same type of calculations that produced Figure 4 and Figure 5 for the case of an MI sensor with *µ*_max_ = 5000 *µ*_0_ have been performed for sensors with *µ*_max_/*µ*_0_ = 500, 100, and 10—that is, with a decreasing intrinsic MI response (intrinsic MI_max_ values are 156%, 46%, and 5.5%, respectively). We introduce a new parameter, *η*, to compare the capacity of the different sensors to detect MNPs with increasing concentrations, defined as
(4)$$\eta \left(\%\right)=\frac{{\mathrm{MI}}_{\max}\left(\mu_{P}\right)-{\mathrm{MI}}_{\max,\mathrm{woP}}}{{\mathrm{MI}}_{\max,\mathrm{woP}}}\times 100,$$
where MI_max,woP_ is the maximum value of the MI ratio in the sensor without particles (the intrinsic MI_max_ value). The parameter *η* therefore quantifies the sensitivity of an MI sensor to the presence of MNPs, expressed as the change in the MI ratio experienced by the sensor when a drop of MNPs is placed on top of it. Figure 6 plots the results obtained for *η* in the sensors with different MI performance (represented by their values of *µ*_max_). Note that the curve for the sensor with *µ*_max_ = 5000 *µ*_0_ is the same as the one plotted in Figure 5. 

The data in Figure 6 is certainly conclusive: the sensors with worse MI performance are more sensitive to the presence of MNPs. If extrapolated, the best sensor material for detecting NMPs should be one with no MI effect at all. This is not completely surprising, as there is experimental evidence of a larger response from a plain Cu sensor than from an MI amorphous ribbon in a measurement performed in the same conditions [27]. In fact, this behavior could have been anticipated: if the change in the impedance of the sensor in the presence of MNPs is a consequence of changes in the distribution of the electromagnetic field due to the permeability of the particles, then this effect should be larger when the permeability of the sensor itself is smaller than that of the system of MNPs. The approach of using the impedance changes of nonmagnetic conductors to quantify the concentration of MNPs is being used successfully in the development of magnetic biosensors [28].

In accepting these conclusions, one should be aware of the limitations of the present analysis. The results from the FEM calculations do not include many relevant physical effects that occur in real systems, such as the dependence of the permeability on the frequency, or the possible agglomeration effects that takes place when the particle concentration is high, which completely change the magnetic behavior of the ensemble. In particular, the latter effect is difficult to model, but probably will modify the aspect of Figure 5 and Figure 6 in which the MI sensitivity continuously increases with an increase in particle concentration.

Finally, it should be reminded that the analysis presented here was made for the case of superparamagnetic nanoparticles that do not produce stray magnetic fields unless magnetized. Highly sensitive magnetic sensors can still be used to detect larger monodomain nanoparticles presenting a net magnetic moment [29]. The modeling of the MI signal produced by ferromagnetically behaving particles requires a different approach to the one used here [30].

## 4. Conclusions

The performance of MI sensors to detect the presence and concentration of magnetic nanoparticles is investigated using finite element calculations to directly solve Maxwell’s equations. The assumption is made that the sensitivity to MNPs comes from their magnetic permeability, and not from the stray magnetic field that they produce. This should be the case for superparamagnetic particles if they are not magnetized by an external magnetic field of sufficient strength. In this hypothesis, the variation of impedance experienced by the MI sensor is a consequence of the change in distribution of the electromagnetic field due to the permeability of the MNPs system.

The results obtained not only confirm that the sensitivity of MI sensors can be justified considering only the permeability of MNPs, but also help explain the origin of the discrepancies found in the literature about the response of magneto-impedance sensors to the presence of magnetic nanoparticles, where some authors report an increasing MI signal when the concentration of MNPs is increased, while others report a decreasing tendency. The results show that the change of the MI response when increasing the concentration of MNPs can be positive or negative, depending on the effective permeability of the particle system.

Additionally, the study demonstrated that a good intrinsic MI performance of the sensor has a detrimental effect on its capacity to detect MNPs: sensors with a lower MI response display larger sensitivities to the presence of magnetic nanoparticles. This seems to confirm that the correct strategy for detecting MNPs is to use the impedance change of plain, nonmagnetic conductors as sensing materials, at least in the case of superparamagnetic particles that are not magnetized in an external magnetic field.

## Figures and Tables

**Figure 1 sensors-20-01961-f001:**
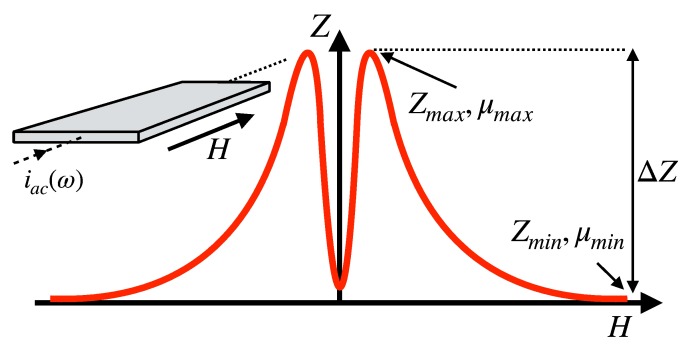
Sketch of the dependence of the impedance on the applied magnetic field in a soft magnetic sample with transverse anisotropy (in-plane easy axis, perpendicular to the current flow and the applied field).

**Figure 2 sensors-20-01961-f002:**
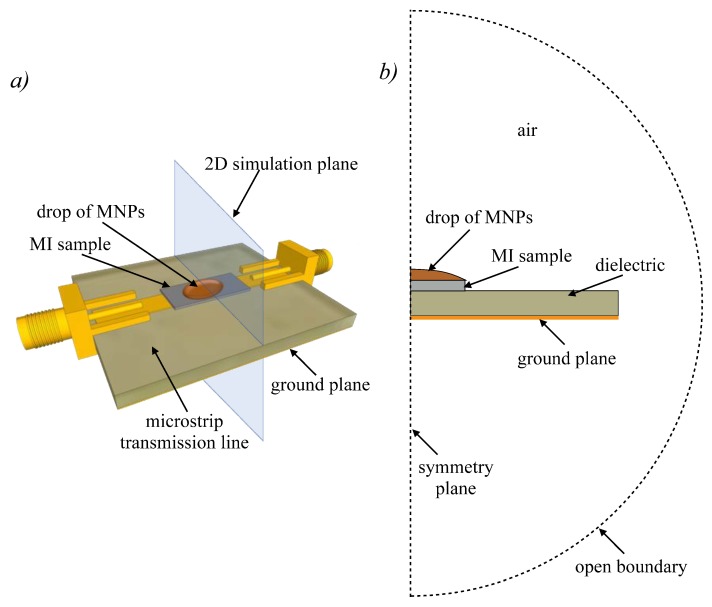
(**a**) Illustration of the experiment simulated by FEM: a drop containing a certain concentration of magnetic nanoparticles (MNPs) is placed on the MI material, which is inserted in a microstrip line to determine its impedance. (**b**) Layout (not at real-scale) of the two-dimensional problem solved by FEM, which corresponds to the middle plane of the setup as indicated in (**a**). Only half of the plane needs to be simulated due to symmetry.

**Figure 3 sensors-20-01961-f003:**
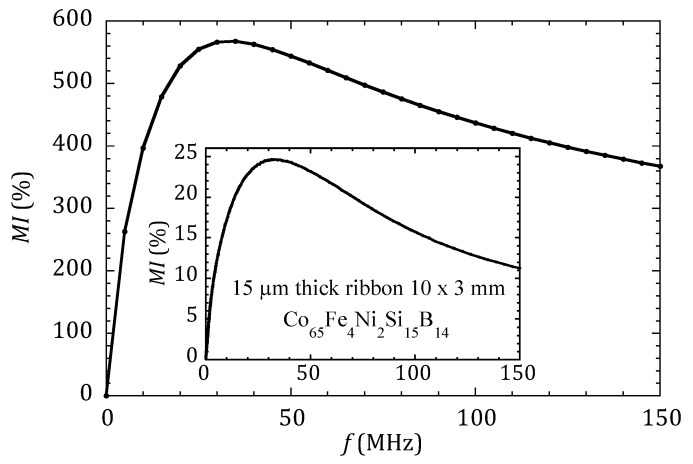
Magneto-impedance ratio MI as defined in Equation (2), calculated using FEM for the case of a planar sample 1 mm wide and 20 µm thick, with the permeability *µ*_max_ = 5000 *µ*_0_. For comparison, the inset shows the MI ratio experimentally measured in an amorphous ribbon.

**Figure 4 sensors-20-01961-f004:**
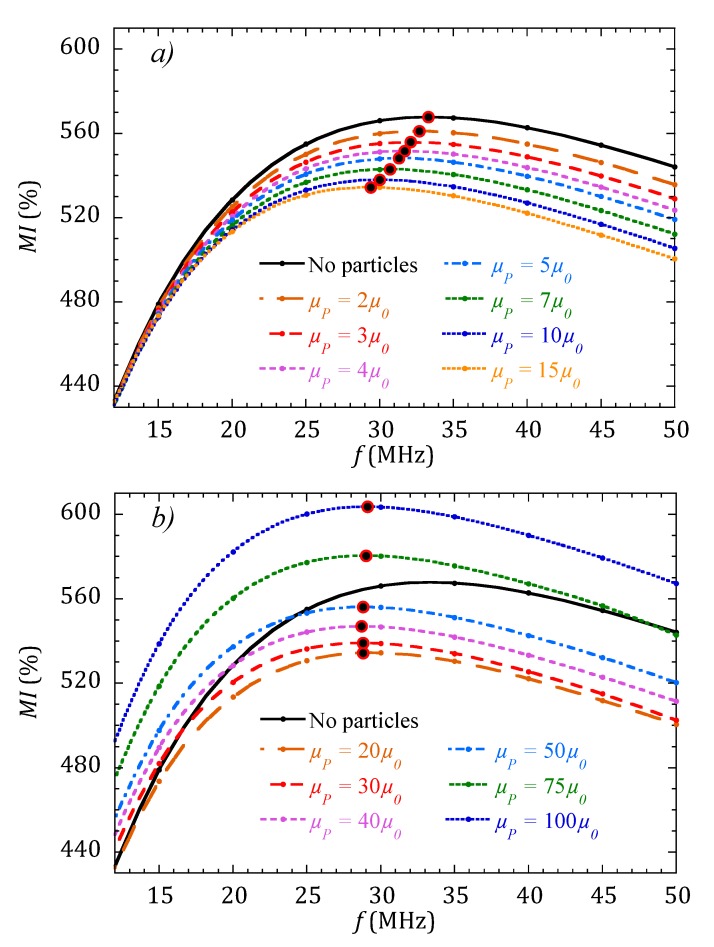
Variation of the MI curves calculated for a sensor with *µ*_max_ = 5000 *µ*_0_ when a drop with increasing MNPs concentration is placed on it. (**a**) for concentrations up to *µ_P_* = 15 *µ*_0_, the magnitude of MI decreases. (**b**) For larger concentrations, from *µ_P_* = 20 *µ*_0_, the magnitude of MI increases. Large red and black dots indicate the position of MI_max_ for each curve.

**Figure 5 sensors-20-01961-f005:**
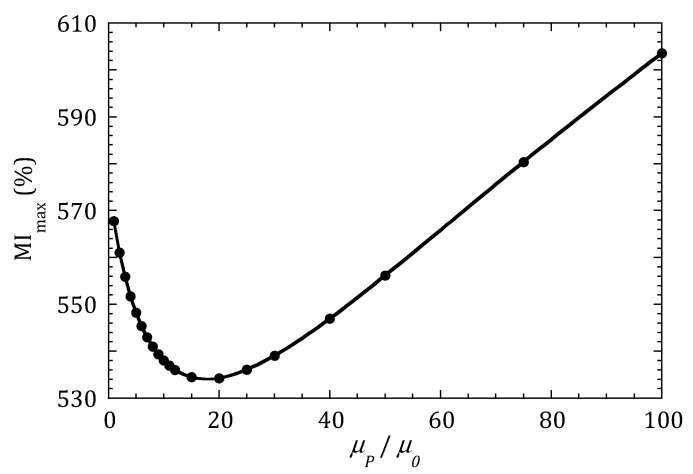
Maximum MI ratio of a sensor with *µ*_max_ = 5000 *µ*_0_ when a drop of MNPs is placed on it, as a function of the concentration of nanoparticles (expressed as increasing permeability *µ_P_* values).

**Figure 6 sensors-20-01961-f006:**
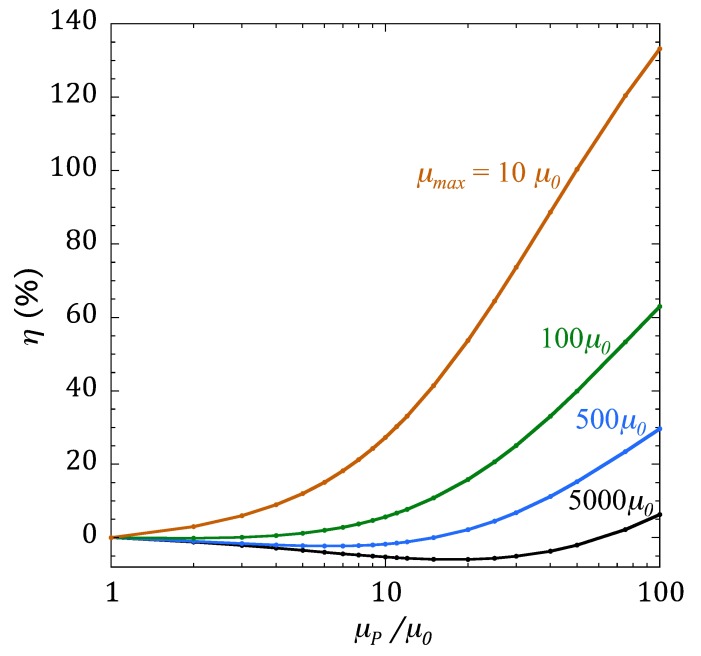
Relative change of the MI_max_ response, experienced by sensors with different MI performance (different values of *µ*_max_) when a system of MNPs with increasing concentration is placed on top.

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
