# Peer review of "The Performance of the Magneto-Impedance Effect for the Detection of Superparamagnetic Particles"

_sensors, 2020, doi:10.3390/s20071961_

Round 1
Reviewer 1 Report
This is an interesting work. The results obtained are valuable and the analysis is sound. The work indeed advances the current understanding of the GMI-based detection of magnetic nanoparticles. I would like to recommend it for publication in this journal, after the author addresses the following comments:
- Since the magnetic nanoparticles (NPs) used in previously reported works could not be purely superparamagnetic, although the authors claimed that. The reason is that these NPs (depending their coating material thickness) can easily agglomerate to form NP clusters that are ferromagnetic (Hc >0, Mr >0) rather than superparamagnetic (Hc = 0, Mr = 0). As a result, these clusters produce stray magnetic fields that would also influence the GMI signal. The conflicting results that indicate either increase or decrease in GMI due to the presence of "superparamagnetic" nanoparticles could also arise from this fact. It is therefore suggested that the author extends discussions to include this point in the manuscript.
- From the calculations, it appears that the particle detection sensitivity keeps increasing with increasing concentration of superparamagnetic NPs. This in fact differs from experiments because at high concentrations, NP clusters are likely formed, thus inducing stray fields that would also have some influence on the GMI signal. Again, the above and this point should be discussed in the presence manuscript.
- It would also be useful if the author could extend discussions on detection of ferromagnetic NPs in this manuscript.
Author Response
Please, see attached document.

Reviewer 2 Report
This paper intends to show using finite element calculations how the magnetoimpedance effect (MI) might be used to detect the presence of magnetic nanoparticles. No details are given of the magnetic system and the result obtained affirms that the sensitivity can be both positive and negative. The author affirms that this behavior is seen in the literature but he does not provide references! The paragraph following Figure 5 would be the ideal place to insert these references. Please do this!
Let's suppose that the behavior shown in Figure 5 is correct. What is the physical reason for the minimum in the curve? What physical parameters determine the permeability for which the minimum occurs and the size of this minimum. Are there cases where the minimum does not appear? The author must provide a complete discussion of Figure 5 as well as references to experimental papers where this phenomenon occurs.
The author mentions that this paper is relevant to work on biosensors. Please explain and provide an example.
Author Response
Please, see the attached document.

Reviewer 3 Report
Overall Recommendation: Minor revision
This paper reports the performance analysis of a magneto-impedance (MI) sensor through numerical simulation. The analysis shows the effect of particle concentration on the magneto-impedance at low frequencies, which describes the phenomenological behavior of the MI sensor in different experiments, but not a quantitative agreement. Therefore, the results appear to be worthy of publication after the author addresses a few minor recommendations.
- The are several unreferenced statements in the introduction. e.g. lines 26 – 29, 38-39, 57, 64, 82, 102, etc. It is recommended to add a reference and to be more explicit when describing the previous work, i.e., who are they? Who are the others? It would help if you referenced them.
- Is there a typo in Eq. (3)? sqrt(j) or j
- The simulation domain is not clearly explained. Are you going to simulate the vertical plane? Or a plane parallel to the ground plane? Show it clearly in Figure 2.The caption in Figure 2 says that you are going to simulate the middle plane, but it is not clear in the figure or the procedure description.
- Please don’t use the acronyms in the abstract and defined them right after you use them.
Author Response
Please, see the attached document.

Round 2
Reviewer 2 Report
The properties of the nanoparticles do not seem to enter into the calculation. In fact there is nothing to indicate that the medium surrounding the MI sensor is composed of nanoparticles. Could it be steel plate? How would the results be changed if the size of the particles were changed? What were the dimensions of the nanoparticles used in this calculation? What was their shape? Describe the FEM calculations and the size of the particles.
Please go into detail about how the calculation would change if the particles were ferromagnetic.
Does figure 2 and the data of figure 3 really indicate that your laboratory has the capacity to make these measurements? If that is the case, then it would greatly improve the paper to show actual measurements.
Author Response
Please, see the attached document.
